# Au-Hyperdoped Si Nanolayer: Laser Processing Techniques and Corresponding Material Properties

**DOI:** 10.3390/ma16124439

**Published:** 2023-06-16

**Authors:** Michael Kovalev, Alena Nastulyavichus, Ivan Podlesnykh, Nikita Stsepuro, Victoria Pryakhina, Evgeny Greshnyakov, Alexey Serdobintsev, Iliya Gritsenko, Roman Khmelnitskii, Sergey Kudryashov

**Affiliations:** 1Lebedev Physical Institute, 119991 Moscow, Russia; 2School of Natural Sciences and Mathematics, Ural Federal University, 620000 Ekaterinburg, Russia; 3Laser and Optoelectronic Systems Department, Bauman Moscow State Technical University, 2nd Baumanskaya St. 5/1, 105005 Moscow, Russia; 4Institute of Physics, Saratov State University, 410012 Saratov, Russia

**Keywords:** amorphous Si film, laser hyperdoping, gold impurity, energy-dispersive X-ray microspectroscopy, X-ray photoelectron microspectroscopy, Raman microspectroscopy, IR spectroscopy

## Abstract

The absorption of light in the near-infrared region of the electromagnetic spectrum by Au-hyperdoped Si has been observed. While silicon photodetectors in this range are currently being produced, their efficiency is low. Here, using the nanosecond and picosecond laser hyperdoping of thin amorphous Si films, their compositional (energy-dispersion X-ray spectroscopy), chemical (X-ray photoelectron spectroscopy), structural (Raman spectroscopy) and IR spectroscopic characterization, we comparatively demonstrated a few promising regimes of laser-based silicon hyperdoping with gold. Our results indicate that the optimal efficiency of impurity-hyperdoped Si materials has yet to be achieved, and we discuss these opportunities in light of our results.

## 1. Introduction

Silicon hyperdoping is intensively studied to extend the IR response for diverse optoelectronic applications [1,2,3]. At present, the hyperdoping of Si (Figure 1) could be achieved by ion implantation and laser processing [4,5], enabling the management of impurity concentration and distribution. The main impurities are nanoparticles of Er, Ag, Fe, Ti, Ni, Zn [1], while hyperdoping by gold (Au) nanoparticles [6,7,8] is less understood exhibiting high impurity mobility in a Si matrix [9]. In the system, Au-Si bonding in the Si matrix is considerably weakened, supporting the diffusion of Si atoms through a gold overlayer and room-temperature elemental mixing in a solid solution [10]. As an acceptor impurity, gold leads to impurity levels near the valance band of Si. During ion implantation, some host Si material and doped Au impurities could be sputtered compared to material- and cost-effective implantation via ultrashort-pulse (femto-, sub- and picosecond) laser processing when hyperdoping at levels in excess of 1 at% (impurity concentration ~5 × 10^20^ cm^−3^ [11], well above the equilibrium solubility limit of 10^15^ cm^−3^ [12]) could be achieved due to extreme laser-heating, melting and solidification (quenching) rates. The optical and electronic properties of gold-ion implanted Si monocrystals due to the produced Au nanoparticles (np), including fabricated NIR photodetectors, were explored in [13,14].

Thin Si films are known to exhibit a diminished heat conduction ability due to the quantum phonon confinement effect, limiting the density of states in the corresponding phonon spectrum—the effect of their being harnessed in flexible solar elements in microelectronics. Wang et al. [15] prepared Ti-hyperdoped amorphous silicon (α-Si) films through magnetron sputtering and laser annealing of the multi-layer set of α-Si and Ti layers on the crystalline Si substrate. These studies resulted in a quite homogeneous distribution of doping Ti in Si at an average concentration of 5 × 10^20^ cm^−3^, providing the 1200 nm absorption coefficient of 1.2 × 10^4^ cm^−1^. Multi-layered Si–S–Si films were prepared by thin-film deposition and nanosecond laser-processing [16], demonstrating an optical absorbance of 90% (75–90% in the NIR spectral range), carrier concentration ~1 × 10^15^ cm^−2^ and carrier mobility of 72 cm^2^/(V·s) in the sulfur-hyperdoped silicon sample. Boron-doped nanocrystalline (nc-Si) 230 nm thick films were grown on Si <100> substrates at 1000 °C, showing a carrier mobility of 15 cm^2^/(V·s) at 300 K [17].

This study was focused on comparing the nano- and picosecond laser hyperdoping/annealing of α-Si film in a liquid CS_2_ environment or air environment and a quite comprehensive chemical, structural and IR characterization of the resulting samples for their potential application in photovoltaics.

## 2. Materials and Methods

First, 530–550 nm thick a-Si films were deposited on substrates using magnetron setups at a chamber pressure of 10^−2^–10^−3^ Pa, target voltage and current of 500–650 V and 1.5–2 A, respectively. The deposition rate was 0.1–0.4 µm/min. As substrates for the studied silicon films, glass slides with conductive layers based on aluminum (200 nm) and chromium (550–570 nm) deposited on them were used to form a rear current-carrying contact. The film thickness was chosen on the basis of theoretical calculations of the distribution of the ranges of gold ions over the depth of the silicon matrix and preliminary experimental studies. Next, a gold film was deposited onto amorphous silicon in a magnetron sputtering facility in an argon atmosphere. Single-pass raster-scanning of 2 × 2 mm^2^ regions of amorphous silicon films with a top gold film (50 nm) was carried out using two workstations for ablative laser processing: (1) comprising an ytterbium-doped fiber nanosecond laser HTF Mark (OKB «Bulat», Moscow, Russia) with central wavelength λ = 1064 nm, maximum output energy E_max_ up to 1 mJ in the TEM_00_ mode, FWHM pulsewidth τ = 120 ns, repetition rate f = 20–80 kHz, and (2) comprising an ytterbium-doped fiber laser Satsuma (Amplitude Systemes, Paris, France) with central wavelength λ = 1030 nm, maximum output energy E_max_ up to 10 μJ in the TEM_00_ mode, FWHM pulsewidth τ = 0.3–10 ps, repetition rate f = 0–500 kHz (Figure 2). The laser beam was focused by a galvanoscanner ATEKO^TM^ (ATEKO, Moscow, Russia) with an objective focal length ≈100 mm. Samples were arranged inside a glass beaker with a 5 mm thick liquid carbon disulfide (CS_2_) top layer or without solvent (in air) for 120 ns pulses and for 10 ps pulses. Then, the sample was scanned across a 2 × 2 mm^2^ square area in a multi-spot pattern with 100 lines/mm filling at repetition rate of 80 kHz in the case of ns-laser processing and 160 kHz in the case of ps-laser processing. Scan velocities for 120 ns and 10 ps pulses were V = 80 mm/s and V = 50 mm/s (surface exposure: N = 40 and 60 shots per spot), respectively.

The laser exposure was provided by 0.1-mJ 120-ns pulses (0.5 µJ for 10-ps pulses) focused onto the sample surface into a spot with the 1/e-radius σ_1/e_ ≈ 20 μm (10 μm for 10-ps pulses), corresponding to peak laser fluences of 8 J/cm^2^ and 0.15 J/cm^2^, respectively. Subsequently, for both ns- and ps-laser processing, two regions were selected, in which the gold concentration was maximum and the oxygen concentration was minimum, respectively. This choice is due to the fact that, when creating a photodetector based on hyperdoped silicon, an important parameter is the absence of defects—micro-cavities, cracks, etc. The less oxygen is contained in the treated material, the less such defects are present. In addition, the maximum concentration of an impurity material (gold) is important for the registration of IR radiation, so we chose the area with the highest content. All subsequent studies were carried out with these two sample regions. Thus, processing in liquid CS_2_ became optimal for ns pulses, and ambient air became optimal for ps pulses.

The surface topography and chemical composition of the nanopatterned spots was characterized by means of a scanning electron microscope (SEM; VEGA, TESCAN, Brno, Czech Republic), equipped with an energy-dispersion x-ray spectroscopy (EDX) module Aztec One (Xplore EDX detector; Oxford Instruments, High Wycombe, UK) for chemical micro-analysis at the 3, 5, 7 and 10 keV kinetic energies of electrons.

Chemical states of gold-doping were studied in the laser-hyperdoped spots of the sample at a high vacuum (~10^−9^ mbar) by means of an x-ray photoelectron spectrometer (XPS) K-Alpha+ (Thermo Fisher Scientific, UK). XPS spectra were acquired from the spots of 400 µm in diameter in the ranges of O 1*s*, C 1*s*, S 2*p* Si 2*p*, and Au 4*f* lines, using an Al-Kα monochromatic source, 50 eV pass energy at the energy accuracy of 0.1 eV; composition accuracy was ≈0.5 at.%. The spectrometer was calibrated to the binding energy of Au 4f_7/2_ line at 83.95 eV. The spectral line shift was corrected in accordance with C 1*s* (284.9 eV) and Au 4*f* (83.95 eV) lines.

The crystalline state of the laser-processed spots of the sample was characterized by room-temperature 3D-scanning confocal Raman/PL microspectroscopy, using a Confotec 350 (SOL instruments, Minsk, Belarus) microscope-spectrometer at a 532 nm laser excitation wavelength with a spectral resolution of 0.5 cm^−1^.

The reflection of laser-processed spots of the sample was finally characterized in ambient air by room-temperature Fourier-transform infrared (FT-IR) (650–2500 cm^−1^) spectroscopy, using a spectrometer FT-805 (Simex, Novosibirsk, Russia) with a spectral resolution of 0.5 cm^−1^. In the wavenumber range from 2500 to 10,000 cm^−1^ the reflection spectrum was obtained using an IR spectrometer VERTEX 70v (Bruker, Karlsruhe, Germany) with a spectral resolution of 0.5 cm^−1^.

## 3. Results and Discussion

Figure 3a–h show SEM images of sample surfaces and EDX elemental maps of silicon, gold, and oxygen after ns- and ps-laser processing, respectively. The analysis of SEM images allows for us to draw conclusions about the topology of the modified surfaces. During ps treatment (Figure 3e), a nanostructure is formed on the surface of the sample; this is absent during ns treatment (Figure 3a) and characterizes the use of ns pulses as a gentler method of laser-processing amorphous silicon. On the other hand, when using ns pulses, separate material clusters were formed on the surface (white spots in the SEM image). In addition, the resulting nanorelief on the surface in the case of ps-laser processing can be interpreted as light-trapping, which significantly reduces the reflection coefficient of the surface in the visible region of the spectrum [18]; however, within the framework of this article, this fact is not an advantage.

EDX elemental maps of silicon and gold show that the ps-laser processing of the region (Figure 3f,g) provides a more uniform distribution of atoms of chemical elements on the surface compared to the ns-laser processing (Figure 3b,c), in which separate gold clusters were formed, taking into account the fact that the EDX study was carried out with the same parameters of the electron beam (accelerating voltage, cathode current, working distance, etc.) for two regions. In this case, the distribution of oxygen atoms is uniform for both cases (Figure 3d,h), although treatment with ns pulses provides a lower concentration of oxygen atoms, which will be quantitatively shown in the results of the EDX study. This difference can be explained by the presence of an oxygen-free environment (CS_2_ liquid) in the case of ns-laser processing. It should be noted elemental maps were also taken for other penetration depths. However, in these cases, the distribution of atoms of the main chemical elements remained uniform everywhere, which means that the above-described irregularities are typical only for the sample surface.

Figure 4 and Figure 5 show the atomic composition of untreated and treated with ns- and ps-laser-processing regions of the sample, depending on the cathode accelerating voltage of the electron beam during the EDX study. Moreover, in all three cases, the concentrations of silicon (Si), gold (Au), oxygen (O), and aluminum (Al) atoms are indicated due to the absence of other chemical elements in the study of modified regions. However, when measuring the concentration of atoms in the untreated region, carbon was also detected in a significant amount, which remained on the surface of the sample after it was immersed in CS_2_; therefore, carbon atoms were not considered in this study.

For a better interpretation of the conducted research, the results were shown as dependences of the concentration of atoms (Si, Au, O, Al) on the accelerating voltage for regions after ns-laser processing (Figure 4) and ps-laser processing (Figure 5). Figure 4 and Figure 5 show the results for the non-modified region. The conversion of the values of the accelerating voltage into an approximate value of the penetration depth at which the atomic composition is studied was carried out according to [19]. The penetration depth values are given on the upper horizontal axis in nanometers.

After the modification, the distribution of Si over the depth for both processing regimes shows a similar character: in the near-surface layer (100–150 nm), the concentration is reduced due to the significant diffusion of gold atoms after irradiation of the regions. Closer to 400 nm, this increases to its maximum value, and after 600 nm there is a decrease in concentration due to the significant increase in aluminum atoms. In this case, the treatment with ns pulses provides a higher Si average concentration by 7.1% compared to ps pulses. The distribution of Au atoms at a penetration depth from 100 nm to 650 nm for the ns-laser processing shows a monotonic decreasing character. The maximum concentration value is reached in the near-surface layer, and the average value is 8.0%. In the case of the ps-laser processing, the concentration of Au atoms remains approximately constant over the entire depth (about 6.6% on average) and the maximum value is reached by 250 nm. The concentration of O atoms for both processing regimes decreases with increasing penetration depth. However, ns-laser processing provides an average oxygen concentration that is lower by 6.3% compared to ps-laser processing. The distribution of Al atoms is approximately the same for both processing regimes—the concentration increases monotonically from 0% to 19% as the penetration depth increases. At the same time, all the above results correlate with the image of the original multilayer structure shown in Figure 2. Thus, according to the results of the atomic composition of the modified sample regions, ns-laser processing is preferable to ps-laser processing due to the higher concentration of silicon and gold atoms, and the lower concentration of oxygen atoms. However, if we consider the uniformity of the distribution of Au atoms over the depth, ps-laser processing shows the best result.

The chemical state of the elements was determined by XPS. Full-range XPS spectra for ns- and ps-laser processing are shown in Figure 6a,b, respectively. All analyzed samples contained O, C, Si, Au and also S for ns-laser processing. A detailed spectrum analysis of the Au 4*f* region (Figure 6a,b insets) showed the presence of gold silicide (Au 4*f*_7/2_ component at 84.9 eV [20]) in the case of ps-laser processing. XPS surface analysis (penetration depth no more than 10 nm [21]) of the main chemical components correlates well with the results of the EDX presented in Figure 4 and Figure 5. Material processing with ns-laser provides a higher gold content and a lower oxygen content compared to ps-laser (Table 1).

To investigate the crystalline properties of the regions after ns- and ps-laser processing, a Raman analysis was performed (Figure 7). The Raman spectra of an amorphous silicon (α-Si) film as a reference sample are also shown in Figure 7. The Raman spectra of α-Si are characterized by two separate bands, at about 170 cm^−1^ and 480 cm^−1^, which are associated with transverse acoustic (TA) and transverse optic (TO) vibrational modes, respectively [22]. After the ns- and ps-laser processing of a silicon sample, a sharp peak appears at about 521 cm^−1^, which corresponds to the crystalline phase of silicon (c-Si). After both processing regimes of laser treatment, an insignificant amount of α-Si is contained. However, after modification with ps pulses, the content of the amorphous silicon phase is higher compared to modification with ns pulses. Such a difference in the crystallinity of the regions after the presented processing regimes is associated with the different durations of the applied laser pulses. In the case of processing silicon with ultrashort laser pulses (fs and ps duration), the the resolidification-front speed after radiation exposure is faster than the relaxation rate of the liquid to crystal, so an amorphous phase of the material appears. Conversely, “long” laser pulses (ns duration) provide a slower resolidification-front speed, so the material acquires a crystalline structure. This effect is well-studied and described in many works, for example, [23]. Thus, according to the above study results, as well as the smoothing of the crystalline peak in silicon [24] on the Raman spectra (Figure 7), it can be judged that silicon after processing appears in the form of polycrystals (poly-Si) [25].

Figure 8 shows the IR reflectance spectra of surfaces—polished crystalline silicon wafers (as a reference sample, as in Figure 8a), unmodified area of the multilayer structure (Figure 8b), treated area after ns-laser processing (Figure 8c) and treated area in ps-laser processing (Figure 8d) in the range of wavenumbers from 650 to 10,000 cm^−1^ (the corresponding wavelengths are indicated on the upper horizontal axis—from 1 to 15 µm). The figure also contains reflectance values after de-coupling the modulation effect to obtain a better understanding of the presented plot (dotted red and blue lines correspond to the spots after ns- and ps-laser processing, respectively). In the plot, the results of measurements of the reflection coefficient from two instruments were combined—from 650 to 2500 cm^−1^ (at wavelengths from 4 to 15 μm), collected using an FT-805 Fourier spectrometer; from 2500 to 10,000 cm^−1^ (at wavelengths from 1 to 4 µm), collected using a VERTEX 70v spectrometer. In all three cases of the reflection coefficient measurement of the unmodified and modified regions (Figure 8b–d), typical oscillations are observed over the studied spectral range. These oscillations are often called spectral interference patterns. Such interference patterns are observed when measuring the transmission or reflection of multilayer structures, and correspond to typical Fabry–Perot modulations [26]. Based on the obtained oscillations, we can estimate the changes in the complex refractive index of the obtained multilayer structure: its real part *n* and the absorption coefficient *k* [25]. The absence of a shift in interference modulations and the invariance of their period in the studied spectral range mean the constancy of the refractive index *n* of the modified region of the material compared to the non-irradiated region. This indicates the preservation of the main material (silicon) of the multilayer structure after processing. In addition, a decrease in the modulation amplitude in the case of the ps-laser processing indicates an increase in the absorption coefficient *k*, which corresponds to the uniform distribution of dopant atoms in the silicon matrix over the depth. Thus, this results also correlate with Figure 5b. Conversely, an increase in the modulation amplitude in the case of the ns-laser processing corresponds to a decrease in the absorption coefficient *k* and some atomic irregularities (Figure 4b). In addition, at wavenumbers of ~450 cm^−1^ (Si-O rocking), ~800 cm^−1^ (Si-O bending), ~1075 cm^−1^ (Si-O stretching), a decrease in reflection was not observed, which indicates an insignificant contribution of the formed surface oxide to the absorption of IR radiation [3]. The minimum values of the reflection coefficient were 7.3% at wavenumber 6164 cm^−1^ (1.6 μm) and 26.7% at wavenumber 1682.2 cm^−1^ (5.9 μm) for the ns- and ps-laser processing, respectively. The average reflection over the studied spectral range was 44.8% for the ns-laser processing and 43.0% for the ps-laser processing. Thus, despite the similar results obtained regarding the average surface reflectance, the ns pulses provide a minimum reflection coefficient of 20% less compared to the ps pulses. In addition, the minimum reflection during ns-laser processing is achieved at wavelength of 1.6 μm, which makes this type of modification preferable for manufacturing IR photodetectors, for example, in telecom applications [27], where the main wavelength of radiation propagating in the optical fiber is about 1.5 µm. On the other hand, ps-laser processing provides an increase in the absorption coefficient, as well as a uniform distribution of gold atoms.

## 4. Conclusions

In this experiment, we were able to hypedope amorphous Si films with gold atoms using ns and ps laser pulses. For each pulse duration, we determined the optimal processing parameters to achieve the highest concentration of metal atoms, exceeding the limit of equilibrium solubility. Further characterization was conducted specifically on these two regions.

At the post-characterization stage, a comparison was made between the silicon surface treatment using ns- and ps-laser processing. The study, conducted using scanning electron microscopy, showed that the ns pulse of treatment is gentler for the silicon surface, as nanoscale structures were found during the ps pulse. Additionally, it was found that, during ns-laser processing, separate clusters of gold atoms were formed on the surface of the sample, although the distribution was less uniform than during ps-laser processing. However, no irregularities were detected in depth. Using energy-dispersive X-ray spectroscopy (EDX), it was shown that treatment with ns pulses provides a higher concentration of gold and silicon atoms, as well as a lower concentration of oxygen atoms compared to ps pulses. On the other hand, ps-laser processing provides a more uniform in-depth distribution of gold atoms. The same results were demonstrated in the study of surface concentration by X-ray photoelectron spectroscopy (XPS). In addition, it was shown that a gold silicide compound was only detected after ps-laser processing. However, after ns-laser processing, gold appeared as inclusions of individual atoms and did not form chemical compounds. The study of crystallinity on a Raman confocal microscope showed that silicon after ns-laser processing is characterized by a higher crystalline phase content than that after ps-laser processing. The reflection spectra of the surface in the range from 600 to 10,000 cm^−1^ (from 1 to 15 μm), revealing Fabry–Perot interference oscillations corresponding to a multilayer structure. The minimum reflection of the surface regions of the irradiated material compared to polished crystalline silicon was reduced by 26.3% and 6.9% for ns- and ps-laser processing, respectively.

By considering the silicon hyperdoping technology used for photoelectronics applications, we found that the critically important parameter that affects the electrical properties of the material is the absence of defects—inclusions of the amorphous phase of the material, oxygen inclusions, and the chemical compounds of silicon with a dopant metal. Research has shown that, when improving the quality of Au-hyperdoping of amorphous silicon, it is necessary to use ns-laser processing, which has a stronger effect than ps-laser processing.

## Figures and Tables

**Figure 1 materials-16-04439-f001:**
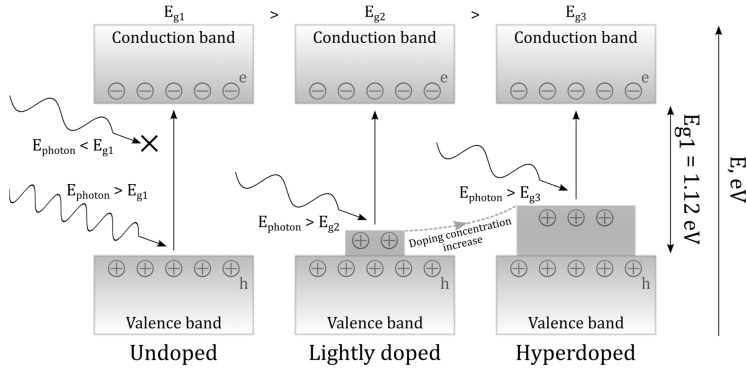
Schematic of Si photoexcitation in undoped (left), doped (center) and hyperdoped (right) cases.

**Figure 2 materials-16-04439-f002:**
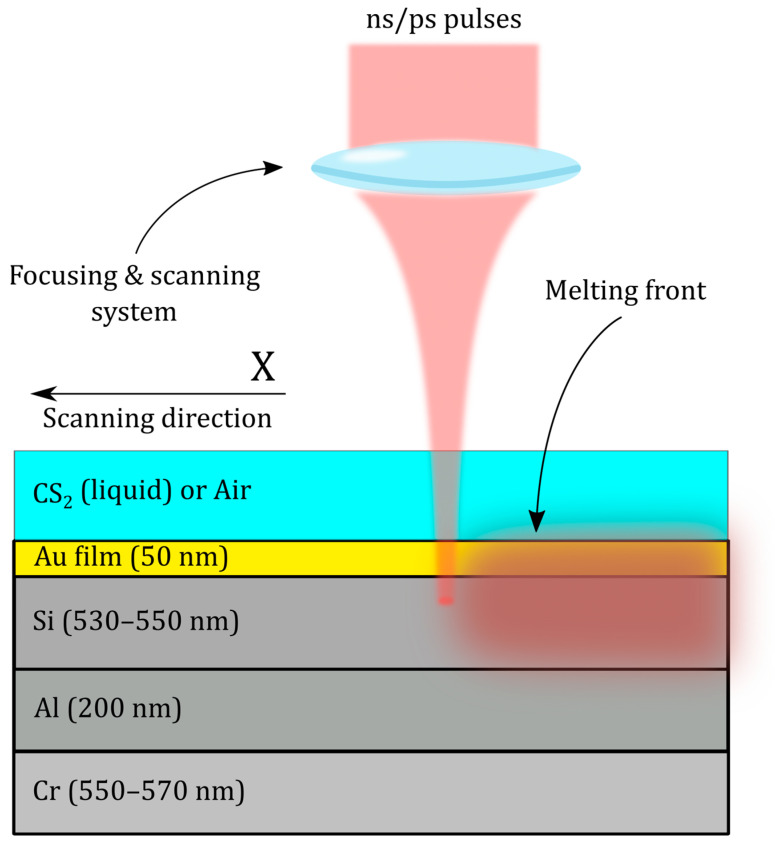
Experimental layout of hyperdoping of a-Si film by its laser melting with a top gold film by nanosecond (ns, CS_2_ environment) and picosecond (ps, ambient air) laser pulses.

**Figure 3 materials-16-04439-f003:**
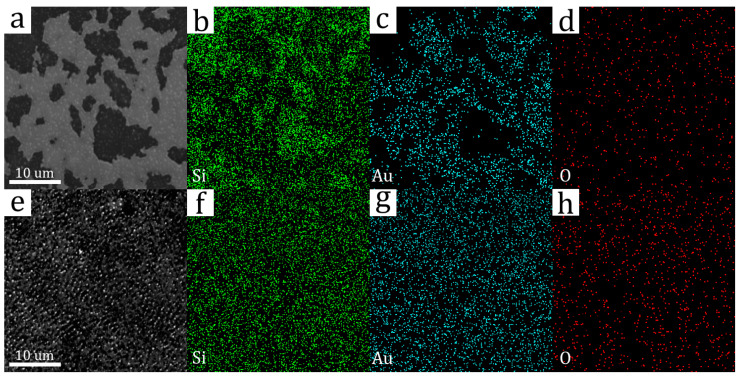
SEM images and EDX elemental maps (Si, Au, O) of sample surfaces after ns- (**a**–**d**) and ps-laser (**e**–**h**) processing.

**Figure 4 materials-16-04439-f004:**
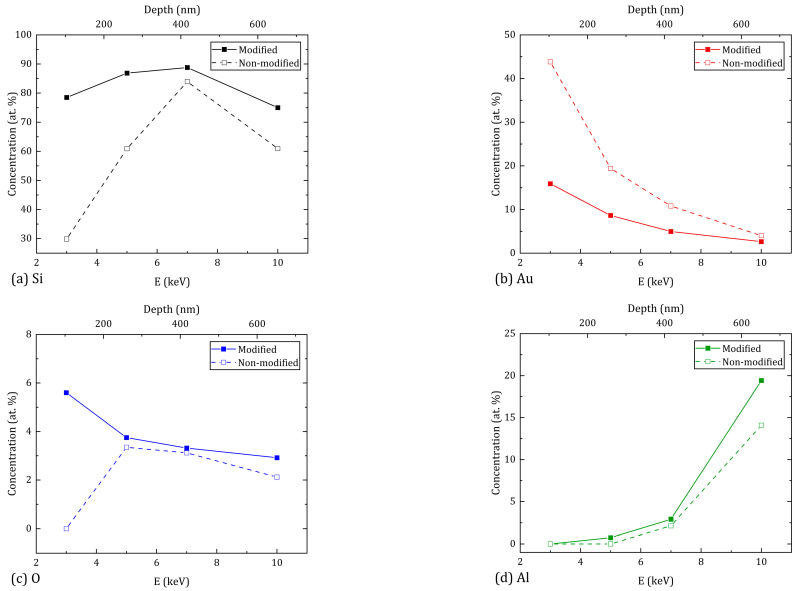
Depth-wise EDX measurements of the main chemical components Si (**a**), Au (**b**), O (**c**), Al (**d**) on the surface of the sample after ns-laser processing. (Error range—0.12%).

**Figure 5 materials-16-04439-f005:**
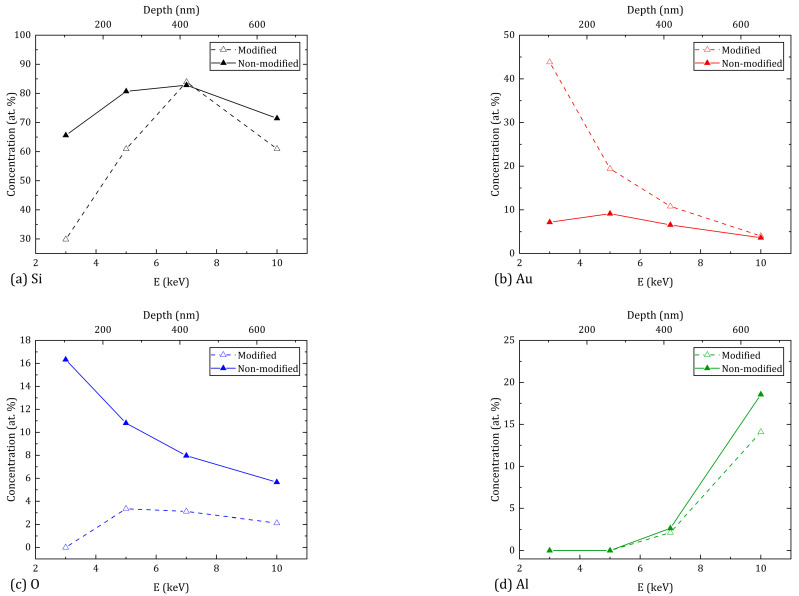
Depth-wise EDX measurements of the main chemical components Si (**a**), Au (**b**), O (**c**), Al (**d**) on the surface of the sample after ps-laser processing. (Error range—0.12%).

**Figure 6 materials-16-04439-f006:**
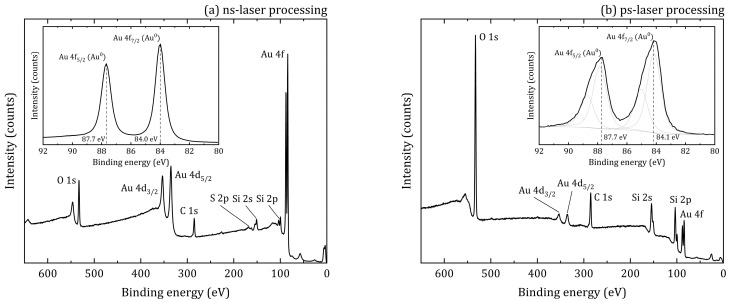
Wide-range XPS spectra and Au 4*f* regions (on the insets) at the surface of the spots after ns- (**a**) and ps-laser (**b**) processing.

**Figure 7 materials-16-04439-f007:**
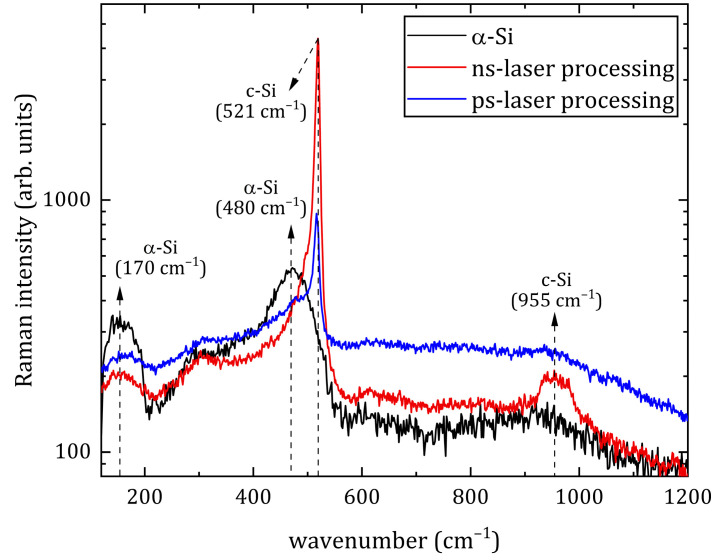
Raman spectra of the α-Si (black line) and ns-laser (red line), ps-laser (blue line) processing regions.

**Figure 8 materials-16-04439-f008:**
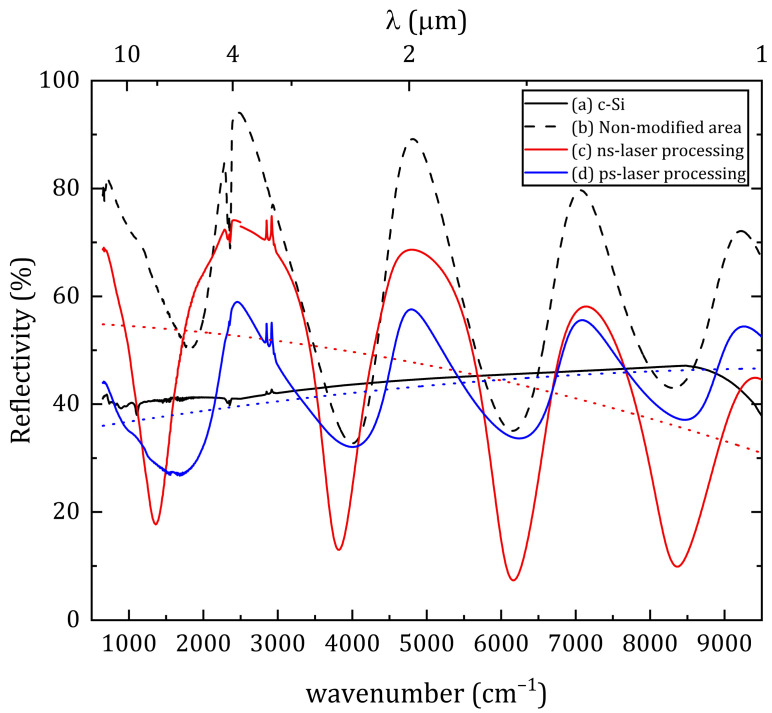
IR reflectivity spectra: (a) Crystalline silicon (c-Si) as reference sample; (b) Non-modified α-Si film with a 100-nm thick Au top layer, (c) ns- and (d) ps-laser processing spots. Dotted red and blue lines correspond to reflectance values of spots after ns- and ps-laser processing, respectively after de-coupling the modulation effect.

**Table 1 materials-16-04439-t001:** Element content determined by XPS for ns- and ps-laser processing.

Element	ns-Laser Processing	ps-Laser Processing
Au	22.4%	1.9%
Si	22.0%	9.8%
Si oxide	14.9%	27.2%
O	38.8%	61.2%
S	1.9%	0.0%

## Data Availability

The supporting data could be provided upon reasonable request.

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
