# Peer review of "Au-Hyperdoped Si Nanolayer: Laser Processing Techniques and Corresponding Material Properties"

_materials, 2023, doi:10.3390/ma16124439_

Round 1
Reviewer 1 Report
The authors applied laser processing techniques and studied properties of Au-hyperdoped Si nanolayer. It is an interesting work in the field of silicon photo detectors. However, there are many flaws in the designation of experiments. For example, nanosecond laser and picosecond laser are used to process materials. Is it mainly to compare the differences in processing quality caused by pulse width (or other parameters)? But this is not a single variable experiment (80 kHz, 80 mm/s for ns laser and 160 kHz, 50 mm/s for ps laser). Figure 4 and Figure 5 are graphical representations of Table 1. It is not recommended to elaborate on the same contents repeatedly. In addition, the resolution of the article images is too low, which affects reading seriously.
Minor editing of English language is required.
Reviewer 2 Report
Kovalev and co-workers have investigated the hyperdoping of Au in amorphous Si thin film using nanosecond and picosecond pulses. Further, they have performed their compositional (EDX), chemical (XPS), structural (Raman), and IR spectroscopic characterization. The authors have thoroughly studied the changes in Si post-irradiation and the amount of Au deposited along with the distribution. Such studies are helpful for preparing materials for NIR applications apart from understanding the later matter (Si in this case) interaction. The experimental work is thorough, and the analysis is adequate. Therefore, I can recommend this work for publication with the following revisions.
- Figure 8: Can the authors plot the reflectance after de-coupling the Fabry-Perot effect? It is difficult to understand the reflectivity of the present plot.
- Resolution of a few figures seems to be poor. Try to improve by providing high-resolution images.
- The authors have mentioned the spot sizes of σ1/e ≈ 20 μm (10 μm for ps pulses) for ns and ps processing. Why was the spot size different? Further, I see that the samples were scanned (raster)? What was the scanning speed? Was it a single-line exposure or multiple lines? All these details need to be provided. Was the repetition rate the same for both pulses?
- Figures 3(a) and 3(e): Can the authors provide better SEM and large-area images?
- Figure 4: The trend for Oxygen as a function of depth for the irradiated film did not follow the unirradiated part, unlike other elements (for both ns and ps cases). Any specific reason?
- Figure 7, Raman data: I expected the ps pulses irradiated film to be more crystalline than the ns irradiated film. Does it have anything to do with the pulse duration, spot size/input fluence, and repetition rate? Further, do the spectra look similar if recorded from different portions of the irradiated part?
Line 269: “we were able to hypedoped amorphous Si films” should be “we were able to hypedope amorphous Si films”
Line 291: “Fabry-Perot interference oscillations corresponds to of a multilayer structure” should be “Fabry-Perot interference oscillations corresponding to a multilayer structure”
Good. Can be improved.
Reviewer 3 Report
The authors conducted an experiment to enhance the efficiency of silicon photodetectors by hyper doping amorphous silicon films with gold using nanosecond and picosecond laser pulses. The study showed that nanosecond laser processing resulted in higher concentrations of gold and silicon atoms, gentler treatment of the silicon surface, and a higher crystalline phase content compared to picosecond laser processing. These findings indicate the potential for improving the quality and performance of gold-hyper doped silicon materials for photoelectronic applications.
To further improve the manuscript, the authors are recommended to address the given comments below:
1. The authors are encouraged to emphasize the novelty of their work and highlight how it surpasses previous studies. Additionally, it would be beneficial for the authors to explain their rationale for selecting 1064 nm and 1030 nm lasers, considering their high reflection on gold in this specific region.
2. The quality of the figures, excluding Figures 2, 7, and 8, needs improvement as they are currently of poor quality.
3. The manuscript would benefit from providing a specific rationale for choosing CS2 liquid. Additionally, for a comparative study, it is important to ensure uniform laser fluence irradiation on both samples. The authors are requested to comment on this aspect.
4. The authors should clarify whether the gold atoms diffuse into the a-Si layer or if they are simply textured with a silicon layer. If gold does indeed diffuse into c-Si, it would be helpful to include cross-sectional EDX analysis, which would clearly demonstrate the diffusion of gold into silicon.
5. The authors may find it a relevant study (https://doi.org/10.1016/j.optmat.2017.05.049), https://doi.org/10.1016/j.jallcom.2023.168818 and similar studies that utilized fs or ps lasers to texture silicon and added thin films of gold nanoparticles. However, it is important for the authors to explicitly explain in what ways their research surpasses such techniques, emphasizing the distinct features and advancements of their own work.
6. Kindly check some typo errors throughout the manuscript, such as line 268# hypedoped
NA
Round 2
Reviewer 1 Report
Thank you for the response. If the processing parameters of testing areas are selected based on the composition contents, please provide a more optimal range of processing parameters to obtain less defects in this study.
Minor editing of English language is required.
Reviewer 3 Report
None
Author Response
There are no Comments and Suggestions for Authors.